# Heptanoate Improves Compensatory Mechanism of Glucose Homeostasis in Mitochondrial Long-Chain Fatty Acid Oxidation Defect

**DOI:** 10.3390/nu15214689

**Published:** 2023-11-05

**Authors:** Siti Nurjanah, Albert Gerding, Marcel A. Vieira-Lara, Bernard Evers, Miriam Langelaar-Makkinje, Ute Spiekerkoetter, Barbara M. Bakker, Sara Tucci

**Affiliations:** 1Department of General Pediatrics, Adolescent Medicine and Neonatology, Medical Centre, Faculty of Medicine, University of Freiburg, 79106 Freiburg, Germanyute.spiekerkoetter@uniklinik-freiburg.de (U.S.); 2Faculty of Biology, University of Freiburg, 79104 Freiburg, Germany; 3Laboratory of Pediatrics, Systems Medicine of Metabolism and Signaling, University Medical Center Groningen, University of Groningen, 9700 RB Groningen, The Netherlandsm.langelaar-makkinje@umcg.nl (M.L.-M.); 4Laboratory of Metabolic Diseases, Department of Laboratory Medicine, University Medical Center Groningen, University of Groningen, 9700 RB Groningen, The Netherlands; 5Pharmacy, Medical Center, University of Freiburg, 79106 Freiburg, Germany; 6G.E.R.N. Research Center for Tissue Replacement, Regeneration & Neogenesis, Department of Prosthetic Dentistry, Medical Center, Faculty of Medicine, University of Freiburg, 79106 Freiburg, Germany

**Keywords:** fatty acid oxidation disorder, glycerol, glucose homeostasis, heptanoate, stable isotope, very long-chain acyl-CoA dehydrogenase, VLCAD deficiency

## Abstract

Defects in mitochondrial fatty acid β-oxidation (FAO) impair metabolic flexibility, which is an essential process for energy homeostasis. Very-long-chain acyl-CoA dehydrogenase (VLCADD; OMIM 609575) deficiency is the most common long-chain mitochondrial FAO disorder presenting with hypoglycemia as a common clinical manifestation. To prevent hypoglycemia, triheptanoin—a triglyceride composed of three heptanoates (C7) esterified with a glycerol backbone—can be used as a dietary treatment, since it is metabolized into precursors for gluconeogenesis. However, studies investigating the effect of triheptanoin on glucose homeostasis are limited. To understand the role of gluconeogenesis in the pathophysiology of long-chain mitochondrial FAO defects, we injected VLCAD-deficient (VLCAD^−/−^) mice with ^13^C_3_-glycerol in the presence and absence of heptanoate (C7). The incorporation of ^13^C_3_-glycerol into blood glucose was higher in VLCAD^−/−^ mice than in WT mice, whereas the difference disappeared in the presence of C7. The result correlates with ^13^C enrichment of liver metabolites in VLCAD^−/−^ mice. In contrast, the C7 bolus significantly decreased the ^13^C enrichment. These data suggest that the increased contribution of gluconeogenesis to the overall glucose production in VLCAD^−/−^ mice increases the need for gluconeogenesis substrate, thereby avoiding hypoglycemia. Heptanoate is a suitable substrate to induce glucose production in mitochondrial FAO defect.

## 1. Introduction

In a healthy state, the cell can efficiently adapt to changing energy demands and availability of nutrients by shifting the use of available substrate [1]. In this process, called metabolic flexibility, mitochondria play an important role, since pathways for NADH, FADH_2_, and ATP production are localized in this organelle [2]. Defects of mitochondrial fatty acid β-oxidation (FAO), such as a deficiency in very long-chain acyl-CoA dehydrogenase (VLCAD), impair metabolic flexibility, as recently reviewed [3]. VLCAD deficiency (OMIM 609575), caused by a mutation in the *ACADVL* gene, is the most common long-chain fatty acid oxidation disorder [4]. Hypoketotic hypoglycemia, cardiac arrhythmias, rhabdomyolysis, and cardiomyopathy are typical symptoms and commonly occur in VLCAD deficient patients during a metabolic crisis [4]. Implementation of newborn screening programs (NBS) worldwide along with dietary management at diagnosis can successfully reduce the incidence of the symptoms [5,6]. 

Due to the increased need of glucose for energy supply in VLCAD deficiency, the liver plays a critical role to maintain glucose and energy homeostasis by regulating endogenous glucose production (EGP) via glycogenolysis and gluconeogenesis [4,7]. Treatment recommendations include avoidance of prolonged fasting, restriction of long-chain fats in the diet, and supplementation with medium-chain triglycerides (MCTs) [8,9]. Recently, the FDA approved the additional use of triheptanoin for the treatment of long-chain fatty acid oxidation (lc-FAO) disorders [10]. Triheptanoin is a triglyceride composed of three heptanoates (C7-acyl groups) esterified to a glycerol backbone. This compound is degraded to propionyl-CoA, which in turn can be converted to anaplerotic substrates for the tricarboxylic acid (TCA) cycle and from there feed into gluconeogenesis [11]. In addition, glycerol itself is also one of the major substrates of gluconeogenesis [12]. A clinical study has shown that triheptanoin is more effective than an MCT diet in reducing the incidence of hypoglycemia [13]. Although triheptanoin did not show a significant effect in preventing cardiomyopathy and rhabdomyolysis, the diet was beneficial in reducing hypoglycemic events in patients with VLCAD deficiency [14]. However, studies investigating the effect of a triheptanoin diet on glucose homeostasis are limited. Labeling metabolites with isotope tracers is a powerful method for measuring the dynamics of metabolic pathways and determining the contribution of various nutrients [15]. In fact, stable isotopes are already used in several clinical diagnoses and in research to dynamically assess in vivo metabolism in the pediatric population [16,17]. In particular, for inherited metabolic disorders, the stable isotope method provides valuable information on metabolic reprogramming and proves to be very specific to the individual patient [18]. 

In the present study, we investigated the role of the liver in the response to defects in mitochondrial FAO using the very-long-chain acyl-CoA-deficient (VLCAD^−/−^) mice as the disease model. We examined EGP with uniformly labeled ^13^C_3_-glycerol, which serves as both a tracer and a gluconeogenic substrate. We also investigated whether triheptanoin, as an alternative gluconeogenic substrate, affects the use of ^13^C_3_-glycerol. In addition, based on the data, we constructed a computational model to quantify the clearance of glucose in VLCAD^−/−^ mice. 

## 2. Materials and Methods

### 2.1. Animal Experiment

The VLCAD^−/−^ and WT control mice were generated from intercrosses of C57BL6+129sv VLCAD genotypes as described previously by Exil et al. [19]. Each group consisted of 10–12 mice, with equal numbers of male and female mice. The mice were given ad libitum access to normal mouse chow. At 12–15 weeks of age, the mice were fasted for 6 h and randomly assigned into four different groups: 1 mmol/kg-BW glycerol; 1 mmol/kg-BW glycerol + 1.5 mmol/kg-BW heptanoate (C7); 1 mmol/kg-BW ^13^C_3_ glycerol; and 1 mmol/kg-BW ^13^C_3_ glycerol + 1.5 mmol/kg-BW C7. The first two groups were denoted non-labeled, whereas the other two were labeled. The group injected with glycerol only or a glycerol-C7 mixture are referred to as control and C7, respectively. The single bolus of glycerol and C7 was administrated by tail-intravenous injection. Ketone bodies were measured at time 0, whereas total blood glucose was measured at time 0, 30, and 60 min after bolus injection using Abbot, FreeStyle Precision (Abbott GmbH, Chicago, IL, USA). For analysis of label enrichment, dried blood spots from the tail tip were collected a Guthrie card at 5, 15, 30, and 60 min after bolus injection. At 60 min after bolus injection, the mice were sacrificed by cervical dislocation. The serum was collected via cardiac puncture at the end of the experiment, and further it was used for free glycerol measurement. The liver was collected, snap frozen immediately and stored at −80 °C until used for further analysis. All animal procedures were performed with the approval of the University’s Institutional Animal Care and Use Committee in Freiburg and was in accordance with the Committees’ guidelines (Approval number 35-9185.81/G-21/016).

### 2.2. Glucose Extraction, Derivatization and Gas Chromatography-Mass Spectrometry (GC-MS)

Glucose was extracted and derivatized as described previously [20]. Briefly, a 6.5 mm diameter of bloodspot from each sample was used. The blood spot was immersed in 50 µL of Milli-Q water and incubated at room temperature for 15 min. Further, 500 µL of ethanol was added and shaken for 45 min. The supernatant (200 µL) was collected after centrifugation for 10 min at 14,000 rpm, 4 °C. The supernatant was dried under a stream of nitrogen at 60 °C. After cooling down, glucose from the supernatant was converted into its pentaacetate derivative by adding 100 µL of pyridine and 200 µL of acetic anhydride, followed by incubation for 30 min at 60 °C and another drying step with stream nitrogen at 60 °C. The dried residue was dissolved in 200 µL of ethylacetate and transferred into an injection vial. All samples were analyzed on an Agilent 9575C inert MSD (Agilent Technologies, Amstelveen, The Netherlands).

### 2.3. Free Glycerol Concentration

Free glycerol in serum samples was measured enzymatically using the Instruchemie kit no. 2913 (Delfzijl, The Netherlands) following the manufacturer’s protocol. 

### 2.4. Metabolite Extraction, Derivatization and GC-MS for TCA Cycle Analysis

The metabolite extraction and derivatization from the liver were performed as already described [21]. Briefly, a 10% homogenate of 100–200 mg of powdered-liver tissue was prepared in ice-cold PBS. 50 µL of the internal standard norleucine (Nle) (0.4 mM in Milli-Q water) was added to 1 mL of homogenate, followed by further homogenization using bead beating and sonication. The whole homogenate was transferred to a screw capped tube, and 1 mL of ice-cold methanol and 2 mL of ice-cold chloroform were added. The homogenate was vortexed for 30 min at 4 °C, followed by centrifugation for 10 min at 2500× *g*, 4 °C. The upper aqueous phase was transferred to a new tube and evaporated under a stream of nitrogen at 37 °C. The dried metabolites were dissolved in 40 µL of methoxamine (MOX) in pyridine solution (20 mg/mL). The sample was incubated for 90 min at 37 °C. After cooling down to ambient temperature, the sample was derivatized at 55 °C for 1 h in 60 µL of MTBSTFA with 1% TBDMS-Cl. The derivatives were centrifuged at 1250× *g* for 10 min at ambient temperature. The clear supernatant was collected into a GC-MS vial with microinsert (APG Europe, Uithoorn, The Netherlands). The calibration standards for TCA intermediates and amino acids were prepared and treated in an identical manner. Concentrations and ^13^C enrichment of TCA cycle intermediates and amino acids were analyzed on an Agilent 7890A GC coupled to an Agilent 5975C Quadrupole MS (Santa Clara, CA, USA) equipped with a CTC Analytics PAL auto sampler (CTC Analytic, Zwingen, Switzerland). 

### 2.5. Metabolic Intermediates Concentration 

The lactate concentration was measured enzymatically. First, the liver sample was deproteinized in 10% (*w*/*v*) of 1 M perchloric acid (PCA). The sample was homogenized, followed by sonification. The homogenate was incubated on ice for 5 min, then centrifuged at 13,000× *g* for 2 min in 4 °C. The supernatant was collected and neutralized by adding ice-cold 2 M of KOH to get the pH of supernatant in the range 6.5–8.0. Further, the supernatant was centrifuged at 13,000× *g* for 2 min in 4 °C. The clean supernatant was collected for the enzymatic lactate measurement. 20 µL of each standard solution and sample supernatant were mixed in 200 µL of glycine/hydrazine buffer (0.5 M glycine, 0.4 M hydrazine, pH 9.0) and 25 µL 25 mM NAD^+^. After the initial absorbance was measured in the in 96-wells plate reader (Agilent Bio Tek Synergy H4) at 340 nm, 37 °C, 100 U/mL of lactate dehydrogenase was added into the mixture. The absorbance was measured for 1 h, at 2 min intervals.

### 2.6. GC-MS Data Processing and Isotope Correction

The isotopologue distributions (M_0_–M_i_) and metabolite concentrations of TCA intermediates and amino acids were calculated from peak areas of the [M-57]+ fragment, as described before [21]. MassHunter Quantitative Analysis software for MS (version B.07.00, Agilent) was used to integrate the peak area of each metabolite of interest. To quantify the absolute metabolite concentration, the peak areas from all isotopologues of a specific metabolite were summed up, divided by the peak area of the internal standard (Nle) and related to the calibration curve. The glucose isotopologue from the blood spot were monitored at the ion *m*/*z* 408–414, corresponding to M_0_–M_6_ mass isotopomer. The measured isotopologues of glucose were corrected using IsoCor [22]. The correction for TCA intermediates isotopologues was performed according to Vieira-Lara [23] method to reliably correct low tracer enrichments.

### 2.7. Label Incorporation

To estimate the fraction of glucose derived from gluconeogenesis relative to glycogenolysis, as well as the contribution of ^13^C glycerol to gluconeogenesis, we followed the method described by Hellerstein et al. [24]. The labeled glucose is considered as a dimer that is formed from two molecules of triose phosphate. The ^13^C enrichment of this gluconeogenic precursor pool, i.e., the fraction of the triose phosphate pool that is ^13^C-labeled, is denoted by *p*. The fraction of glucose derived from glycogen is denoted by *F_glycogen_* and the fraction of glucose derived from gluconeogenesis by FGNG. The subfractions of M+0 glucose, M+3 glucose, and M+6 glucose within the gluconeogenesis fraction are denoted by *P*_0_, *P*_3_, and *P*_6_, respectively, with:(1)$$P_{0}={\left(1-p\right)}^{2}$$
(2)$$P_{3}=2\cdot p\cdot \left(1-p\right)$$
(3)$$P_{6}=p^{2}.$$

The factor 2 in *P*_3_ derives from the fact that M+3 glucose can be formed in two ways from a labeled and a non-labeled triosephosphate. It follows that: (4)$$P_{3}/P_{6}=2\cdot p\cdot (1-p)/p^{2}=2\cdot (1-p)/p.$$

Therefore:(5)$$p=\frac{2}{P_{3}/P_{6}+2}.$$

We denote the measured fractions M+0 glucose, M+3 glucose, and M+6 glucose by *Q*_0_, *Q*_3_, and *Q*_6_, respectively. Since M+3 glucose and M+6 glucose are produced only via gluconeogenesis, the ratio $P_{3}/P_{6}$ equals the measured ratio $Q_{3}/Q_{6}$. Therefore, *p* can be calculated directly from the measured data at each time point, based on:(6)$$p=\frac{2}{Q_{3}/Q_{6}+2}.$$

Subsequently, *P*_0_, *P*_3_, and *P*_6_ are calculated by substituting *p* into Equations (1)–(3). Using that *F_glycogen_* and *F_GNG_* sum up to one, the measured fractions *Q*_0_, *Q*_3_ and *Q*_6_ relate to *F_GNG_* via:(7)$$Q_{0}=F_{glycogen}+F_{GNG}\cdot P_{0}=(1-F_{GNG})+F_{GNG}\cdot P_{0}$$
(8)$$Q_{3}=F_{GNG}\cdot P_{3}$$
(9)$$Q_{6}=F_{GNG}\cdot P_{6}.$$

It follows that:(10)$$F_{GNG}=Q_{3}/P_{3}=Q_{6}/P_{6}$$
(11)$$F_{glycogen}=1-F_{GNG}.$$

Here, we assumed that unlabeled blood glucose is derived from glycogenolysis and gluconeogenesis. We have to acknowledge that the unlabeled blood glucose pool also contains pre-existing (endogenous) glucose. The impact of this contribution depends on the turnover (kinetic constant) of blood glucose that will be obtained from computational modelling in this study. 

### 2.8. Computational Modelling of ^13^C Glucose Time Courses

To determine the apparent kinetic constants of blood glucose turnover, we adapted the system of ordinary differential equations (ODEs) model described by Viera-Lara [23] for our experimental setup. The original model was developed for an oral glucose tolerance test with label, building on earlier models by Dalla Man et al. [25], and validated for both humans and mice [23,26]. In the present study, the label comes from gluconeogenic conversion of ^13^C_3_-glycerol into glucose. The injection of ^13^C_3_-glycerol (M+3) produced two types of blood glucose isotopologues, namely M+3 and M+6 glucose. Here, we only used the measured M+3 glucose data in the model. The model consists of three reaction rates (mM·min^−1^) as follows:(12)$$v_{1}\left(c_{1}\right)=k_{1}\cdot c_{1}$$
(13)$$v_{L}\left(c_{1}\right)=k_{L}\cdot c_{1}$$
(14)$$v_{2}\left(c_{2}\right)=k_{2}\cdot c_{2}.$$

Here, *v*_1_ describes the rate of conversion of ^13^C_3_-glycerol (M+3) into ^13^C glucose. The reaction rate of conversion of ^13^C_3_-glycerol into other fates is denoted by *v_L_*. The flux *v*_2_ represents the reaction rate of utilization of blood glucose by peripheral tissues. The rate constants of the corresponding reaction rates are denoted by *k_i_* (min^−1^). We denoted *c*_1_ (mM) and *c*_2_ (mM) as the concentration of labeled blood glycerol and labeled blood glucose, respectively. Only the latter (*c*_2_) was actually measured. Accordingly, the ODEs describing the dynamics of the labeled metabolite pools are:(15)$$\frac{dc_{1}}{dt}=-\left(k_{1}+k_{L}\right)\cdot c_{1}$$
(16)$$\frac{dc_{2}}{dt}=k_{1}\cdot c_{1}-k_{2}\cdot c_{2}.$$

By combining Equations (15) and (16), we can solve the ODEs analytically:(17)$$c_{2}(t)=C\cdot \left(e^{-k_{2}t}-e^{-k_{u}t}\right)$$
in which: (18)$$k_{u}=k_{1}+k_{L}$$
and *C* is a constant that depends on the fraction of glycerol converted to glucose, the apparent rate constant of glycerol utilization *k_u_*, and the initial concentration of labeled glycerol. Although all three constants *C*, *k*_2_, and *k_u_* might, in principle, be identifiable from the time courses; this was in practice not the case, due to a lack of time points. Since we were mainly interested in *k_u_* and *k*_2_, we followed the procedure described by Viera Lara et al. [23]. Based on the data of all mice and aiming for the lowest AIC (Akaike’s Information Criterion for model selection), *C* was fixed at 0.75 mM for all mice. Subsequently, *k_u_* and *k*_2_ were fitted to the time course data of each mouse separately. 

### 2.9. Quantification and Statistical Analysis

Data are represented as mean ± standard error of the mean (s.e.m.). The significant effect of both genotype and C7 injection, as well as its interaction were measured using a two-way ANOVA test. Otherwise stated, the significant differences of C7 injection between genotypes were evaluated using multiple unpaired Student’s *t*-tests with Holm–Sidak’s method. All the statistical analysis was conducted on GraphPad Prism 9.3.1 (GraphPad Software, San Diego, CA, USA). Values were considered significant if adjusted *p*-value (*q*) < 0.05. 

## 3. Results

### 3.1. ^13^C_3_-Glycerol Bolus to Evaluate Gluconeogenesis

We adapted the procedure by Kalemba et al. [27] of using a single bolus injection of stable-isotope labeled glycerol. The dried blood spots at different time points (Figure 1A) were used to evaluate the dynamics of label incorporation from ^13^C_3_-glycerol into blood glucose using computational modelling (Figure 1B). Glucose can be considered a dimer of dihydroxyacetone phosphate (DHAP) and glyceraldehyde 3-phosphate (GAP). Using universally labeled ^13^C glycerol, we anticipated to find three-carbon labeled (M+3) and six-carbon labeled (M+6) glucose isotopologues in blood (Figure 1C). 

### 3.2. Loss of ACADVL Gene Does Not Affect Glucose Homeostasis

First, we measured the blood glucose concentration at different time points in WT and VLCAD^−/−^ mice, in the absence and presence of C7. According to a two-way ANOVA test, there was no significant effect of the genotype or of C7, nor of the interaction between these factors on blood glucose concentration. Independently, we compared the effect of C7 injection in each genotype using multiple unpaired Student’s *t*-test. At time point 0, WT mice that had received a C7 bolus, showed a lower blood glucose concentration than WT mice without C7 treatment (Figure 2A). This result was unexpected because no treatment had been given yet at this time point. Otherwise, we did not observe any significant difference in blood glucose concentration between WT and VLCAD^−/−^ mice, at any time point and regardless of the treatment. To further compare the effect of genotypes and C7 administration, we calculated the area under the curve (AUC) in each group. There was no significant effect of either genotype or C7 on the AUC of the blood glucose concentration (Figure 2B). The concentrations of ketone bodies (Figure 2C) and glycerol (Figure 2D) were also similar among the groups. These results indicate that despite the loss of VLCAD, the mice maintain glucose and ketone body homeostasis.

### 3.3. The Effect of Heptanoate on the Genotype-Specific Contribution of Glycerol to Gluconeogenesis 

Further, we measured the enrichment of M+3 and M+6 ^13^C-glucose in the blood. Both the time-course and the AUC showed that without the C7, M+3 glucose fractional enrichment was significantly higher in VLCAD^−/−^ mice than in WT mice (Figure 3A). With the C7 bolus, the difference between WT and VLCAD^−/−^ disappeared. The M+3 glucose enrichment was almost identical between the control and the C7 group in WT mice at all time points. The M+6 labelling in blood glucose was less than 1%. Although the method was sensitive enough to measure a clear time profile, we did not observe any significant difference in either the time course or AUC of M+6 glucose fractional enrichment (Figure 3B). 

Using the M+3 and M+6 fractions of blood glucose, we estimated the fractional enrichment *p* of M+3 labeled hepatic triose phosphate (DHAP and GAP in Figure 1C), which is the presumed precursor of labeled glucose produced via gluconeogenesis (Figure 3C and Methods) [24]. At early time points, this precursor enrichment was 30–50%, depending on the group, and it declined to 10–20% at later time points. This decline indicates that the contribution of the labeled glycerol decreased in time, clearly reflecting the transient effect of the ^13^C bolus. The non-labeled fraction of the triose phosphate pool probably arises from endogenous, non-labeled glycerol, and other gluconeogenic precursors, such as amino acids or lactate. In the triose phosphate pool enrichment, no significant differences could be observed between WT and VLCAD^−/−^ mice, regardless of the treatment (Figure 3D). In contrast, the C7 significantly decreased the precursor pool enrichment in the WT, suggesting the use of C7 as an alternative gluconeogenic precursor. In the VLCAD^−/−^ mice the C7 had no significant effect. Next, based on the measured glucose enrichment (*Q_i_*), we calculated the fractions of glucose that were derived from gluconeogenesis (*F_GNG_*). Without C7, the fraction of glucose produced by gluconeogenesis was 66% higher in VLCAD^−/−^ mice than in WT mice (Figure 3E, *q* = 0.06). Upon C7 supplementation, this difference disappeared, due to an increase of the contribution of gluconeogenesis in the WT. It should be noted that *F_GNG_* was derived from the total blood glucose subtracted by the fractions of glucose that were derived from glycogenolysis (*F_Gly_*) (Methods). These results indicate that loss of VLCAD induces the conversion of triose phosphate into EGP, which gluconeogenesis as the preferred pathway rather than glycogenolysis. These results may be associated with a significant downregulation of liver glycogen synthetase (GS) protein expression in VLCAD^−/−^ mice (Appendix A). 

### 3.4. Blood Glucose Turnover Is Not Affected in VLCAD^−/−^ Mice

Based on an adapted computational model for an oral tolerance test with label (Viera-Lara [25] and Methods), we calculated the apparent rate constants of the lumped reactions involved in ^13^C_3_ glycerol conversion to glucose and peripheral glucose consumption in this study. After administered in the blood, labeled glycerol was converted by liver enzymes into labeled glucose, also in the blood compartment. This is denoted by rate constant *k*_1_. Part of the ^13^C_3_-glycerol is metabolized for other purposes, denoted by *k_L_* for ‘loss’. (Figure 4A and Methods). The utilization of blood glucose is characterized by rate constant *k*_2_. To evaluate adequacy of the computational model, the measured data should be consistent with the model [28]. We fitted the time course of the M+3 glucose data to this model (Equation (17)). As shown in Figure 4B, the model fitted the measured data well in all experimental groups, demonstrating that is a good representation of the system. The period of 60 min during which label incorporation was followed, was sufficient to cover most of dynamics. Yet, an additional 120 min timepoint could have been informative. Since we have no information about the loss term, we could not differentiate between the fraction of ^13^C_3_-glycerol that was used for gluconeogenesis (*k*_1_) and the fraction used for other metabolic pathways (*k_L_*), such as glycolysis or triglyceride synthesis. Instead, we summed the rate constant of the two different processes into a single rate constant of ^13^C_3_-glycerol utilization (*k_u_*). There were no significant effects of genotype and C7 on the rate constants *k_u_* and *k*_2_ (Figure 4B,C). This suggests that the loss of VLCAD does not affect the overall utilization of glycerol (*k_u_*) nor the glucose utilization by peripheral tissues (*k*_2_).

### 3.5. Effect of C7 on ^13^C Incorporation Central Carbon Metabolism Is Genotype Dependent 

Subsequently, we evaluated whether the genotype and the administration of C7 affected the concentrations of liver metabolites. A two-way ANOVA test shows that the C7 injection significantly affected the aspartate (Figure 5A) and citrate (Figure 5E) concentrations in the liver. In addition, a significant genotype effect was observed in liver citrate (Figure 5E) and succinate (Figure 5I) concentration. There was no statistically significant interaction effect in those TCA cycle intermediate concentration. Thus, regardless of substrate supplementation, the loss of VLCAD increases the liver citrate and succinate concentration.

In order to further investigate the contribution of glycerol and heptanoate as a fuel or anaplerosis substrate in gluconeogenesis, we analyzed the ^13^C enrichment of liver metabolites after injection of ^13^C_3_-glycerol and heptanoate. Despite a very low ^13^C enrichment in these metabolites, the ^13^C_3_ corrected data (Appendix A and Figure 6) confirmed a clearly detectable ^13^C enrichment from the ^13^C_3_ glycerol. Pyruvate (Figure 6A), alanine (Figure 6B), and lactate (Figure 6C) showed a similar pattern of M+3 enrichment, reflecting the rapid and reversible conversion of pyruvate into lactate and alanine. Without the C7 bolus, all three metabolites were significantly higher enriched in VLCAD^−/−^ mice than in WT mice. Moreover, in the presence of C7, the ^13^C enrichment decreased significantly in VLCAD^−/−^ mice for these metabolites, but not in WT. A similar pattern was also shown in the enrichment fraction of TCA cycle intermediates, particularly ^13^C enrichment of M+2 malate (Figure 6D), and M+2 fumarate (Figure 6F). These data suggest that heptanoate indeed serves as an alternative substrate, thereby diluting the enrichment from ^13^C_3_-glycerol in the measured metabolites, as was observed in VLCAD^−/−^ mice. 

The TCA cycle intermediates can incorporate label from ^13^C_3_-glycerol via two routes. Pyruvate carboxylase catalyzes the anaplerosis reaction of pyruvate to oxaloacetate, resulting in M+3 labeled intermediates in the TCA cycle. Alternatively, citrate synthase catalyzes the condensation reaction of acetyl-CoA and oxaloacetate, which results in the formation of M+2 citrate. In general, we observed very low levels of ^13^C enrichment. Notably, the ^13^C enrichment of M+3 in TCA cycle intermediates were almost undetectable (Appendix A), likely due to the low amount of ^13^C-glycerol injected. 

We showed that both the genotype and the C7 bolus affected the M+1 citrate enrichment (Appendix A). The pattern was qualitatively the same as that of M+2 TCA cycle intermediates, i.e., elevated label in the VLCAD^−/−^ group, and normalization to WT level when C7 was given. However, as depicted in Figure 5A no effect was observed in M+2 enrichment of citrate. At least two full cycles of the TCA cycle are required to form the M+1 isotopologue of citrate [29]. One labeled-carbon atom in α-ketoglutarate could be lost into CO_2_ by α-ketoglutarate dehydrogenase. Subsequently, M+1 will be incorporated in TCA cycle intermediates. There was no significant difference in M+1 enrichment in other measured TCA cycle intermediates among the groups (Appendix A). Together, these data support previous data in this study that VLCAD^−/−^ mice have rapid heptanoate metabolism, resulting in ^13^C enrichment dilution in TCA cycle intermediates.

## 4. Discussion

To date, patients with VLCAD deficiency rely on dietary management to avoid clinical symptoms, including hypoglycemia. However, it is not known exactly how the patients utilize gluconeogenic substrates to maintain glucose homeostasis. Using an animal model of the disease, we showed that a loss of the VLCAD protein increased the contribution of gluconeogenic substrates to the production of glucose by the liver. The loss of VLCAD also enhanced the conversion of glycerol into glycolytic and TCA cycle intermediates. Both effects were lost in the presence of heptanoate, suggesting that this is used as an alternative gluconeogenic and oxidative substrate. 

In the present study, we have chosen a fasting period over 6 h to enable the investigation of gluconeogenesis in VLCAD^−/−^ mice; however, avoiding the metabolic derangement as the consequence of energy deficiency. We showed that VLCAD^−/−^ mice maintained glucose homeostasis after 6 h of fasting and showed a higher proportion of EGP from gluconeogenesis than WT mice, strongly suggestive of a compensatory mechanism of glucose homeostasis due to the mitochondrial lc-FAO defect. However, this adaptive mechanism is not able to cope with the energy requirements during intensive stress such as overnight fasting alone or prolonged fasting associated to cold exposure which leads to severe hypoglycemia in different murine models of VLCAD deficiency [28,29]. 

In early time of fasting, glycogenolysis is the main route for EGP [7]. Hepatic glycogen affects net EGP as well as gluconeogenesis proportion of EGP [30]. Our study suggested that loss of VLCAD may alter the hepatic glycogen synthesis (submitted). In line with this, long-chain acyl-CoA dehydrogenase deficient (LCAD^−/−^) mice significantly repress the hepatic glycogen stores [31]. Thus, the absence of glycogenolysis contribution to EGP may provoke hypoglycemia in VLCAD^−/−^ mice during metabolic derangement.

The computational model applied in this study indicates that high dependence on glucose in peripheral tissues of VLCAD^−/−^ mice [32] may not play the main role as compensatory mechanism of glucose homeostasis, as illustrated by the rate constant *k*_2_, which remained identical in all groups. We emphasize that we adapted a model that was originally constructed and validated for oral glucose tolerance tests [23]. However, the difference between the models resided only in the production/absorption of glucose, not in the consumption by peripheral tissues, represented by *k*_2_, in which we are interested here. In contrast, the availability of gluconeogenic substrate appears to be critical in VLCAD^−/−^ mice. LCAD^−/−^ mice develop fasting-induced hypoglycemia due to shortage in the supply of gluconeogenic precursor, specifically alanine [30]. The hypoglycemic risk in VLCAD^−/−^ is much lower than in LCAD^−/−^ mice [33], nevertheless our results do suggest that heptanoate is efficiently used as an alternative gluconeogenic substrate, particularly in the mutant, implying that in principle it could be beneficial in preventing hypoglycemia [15]. 

One of the surprising results is that VLCAD^−/−^ mice have higher concentrations of TCA cycle intermediates in the liver. Mutations in *ACADVL* gene result in a defective β-oxidation with the subsequent decreased production of acetyl-coenzyme A (acetyl-CoA) [34]. Acetyl-CoA works as an allosteric activator of gluconeogenesis and is an essential metabolite for maintenance of the TCA cycle [29]. The steady state concentration of TCA cycle intermediates is tightly regulated by the balance between anaplerotic and cataplerotic flux [29,35]. Thus, loss of VLCAD may result in either inhibition of anabolism to reduce cataplerotic flux, or accumulation of precursors to synthesis of TCA cycle intermediates due to acceleration of anaplerotic flux. The former may not be the case, as VLCAD^−/−^ mice do not affect the total fatty acid concentration in liver and heart [36]. In line with this study, the increased contribution of gluconeogenesis (FGNG) in the VLCAD^-/-^ mouse compared to the WT, in combination with an unchanged blood glucose concentration, suggests that gluconeogenesis is not limited by a lack of acetyl-CoA. Considering the reduction of amino acids availability in LCAD^−/−^ mice [30], we postulate that the elevation of liver TCA cycle intermediates concentration in VLCAD^−/−^ mice is likely due to overactivation of anaplerotic flux via amino acids catabolism. This effect has been reported in people with nonalcoholic fatty liver disease (NAFLD), in which the flux of oxidative TCA turnover is increased [37,38]. 

Heptanoate in the form of triheptanoin is already applied in the treatment of VLCAD deficiency [14]. In patients with lc-FAOD, triheptanoin has been demonstrated to reduce the number of hospitalization and hypoglycemic crisis [14,39]. In VLCAD^−/−^ mice triheptanoin significantly increased liver glycogen in both WT and VLCAD^−/−^ mice compared with normal-chow diet (unpublished). We suppose that a triheptanoin based diet provide glycogen precursors through the formation of glucose 6-phosphate [40]. In humans, liver glycogen is used as energy source during prolonged endurance exercise [41]. Therefore, supplementation with triheptanoin diet in situations of high catabolic stress such as prolonged fasting may prevent or at least reduce hypoglycemic events in patients with VLCAD deficiency. 

In this study, we could not distinguish the fraction of C7 metabolized to acetyl-CoA and propionyl-CoA. To fully evaluate the effect of heptanoate on the TCA cycle intermediates, the contribution of endogenous propionyl-CoA molecules should be also considered. The use of an appropriately labeled heptanoate substrate will allow to quantify the distinct contributions of acetyl-CoA and propionyl-CoA from heptanoate. The use of triheptanoin with either a labeled glycerol or a labeled heptanoate moiety will give insight into the mechanism underlying the anaplerotic effect of the triheptanoin diet.

The ^13^C_3_-glycerol incorporation into glycolytic metabolites (pyruvate and lactate) takes place via the triose phosphate pool [42]. In this study, although the inferred ^13^C enrichment fraction of the triose phosphate pool is similar in the different genotypes, the ^13^C enrichment fraction of pyruvate and lactate are higher in VLCAD^−/−^ mice than in WT mice, indicating a higher dilution of the hepatic pyruvate and lactate pool through unlabeled lactate influx in WT mice. The result corresponds with a study in which mice with a liver-specific deficiency in mitochondrial long-chain fatty acid β-oxidation (Cpt2L^−/−^) were examined [43]. This mutant results in a marked reduction in lactate flux and consequent suppression of blood and liver glucose concentrations [43]. This may be due to the reduction in blood lactate levels as is also observed in LCAD^−/−^ mice [30]. 

We followed the study of Kalemba et al. [27] as they described the method of using a isotope-labeled glycerol tracers in a single bolus injection. We obtained an value of the enrichment fraction of blood glucose that is comparable with other studies using a similar isotope tracer concentration (~1 mmol/kg) [44]. However, we recognize that to evaluate the ^13^C enrichment of TCA metabolites, higher concentration of isotope tracer would allow more detailed analysis of different isotopologues. Studies investigating ^13^C enrichment of TCA intermediates in vivo often use higher concentrations of the isotope tracer than 1 mmol/kg. For example, in one study, a continuous intravenous infusion of ^13^C_3_-lactate (0.160 mmol/kg + 0.040 mmol/kg/min) was used for 4 h [45], corresponding to almost 10-fold higher concentration of the isotopic tracer than we used in this study. Nevertheless, we have shown that a low concentration of ^13^C_3_-glycerol is sufficient for analysis of gluconeogenesis, which is an advantage in translating this approach to human studies.

## 5. Conclusions

In summary, our animal study provides an insight into the possible mechanism of triheptanoin for maintaining glucose homeostasis during dietary management in patients with lc-FAO disorder. In VLCAD^−/−^ mice the glycerol supplemented with diet, as component of triheptanoin oil, is used as gluconeogenic substrate to maintain glucose homeostasis. However, heptanoate also acts as a gluconeogenic substrate as it competes with glycerol by reducing its contribution in endogenous glucose production. In addition, heptanoate seems also to be able to improve TCA efficiency.

Further studies with appropriate combinations and concentrations of isotope tracers are necessary to elucidate whether heptanoate provides additional acetyl-CoA (allosteric activator) or propionyl-CoA (anaplerotic substrate). 

## Figures and Tables

**Figure 1 nutrients-15-04689-f001:**
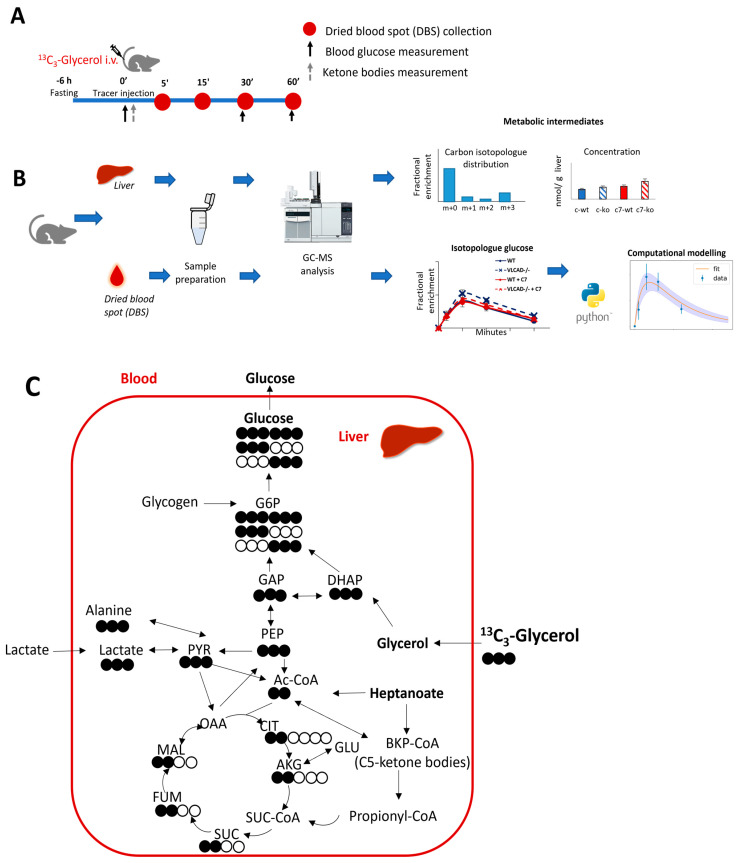
Schematic representation of the experimental design with ^13^C_3_-glycerol as stable isotope tracer. (**A**) Animal experimental design. (**B**) Sample analysis and data generation (**C**). Simplified ^13^C incorporation from ^13^C_3_-glycerol in gluconeogenesis and TCA cycle. Filled and unfilled circles illustrate the labeled and unlabeled carbon atoms, respectively.

**Figure 2 nutrients-15-04689-f002:**
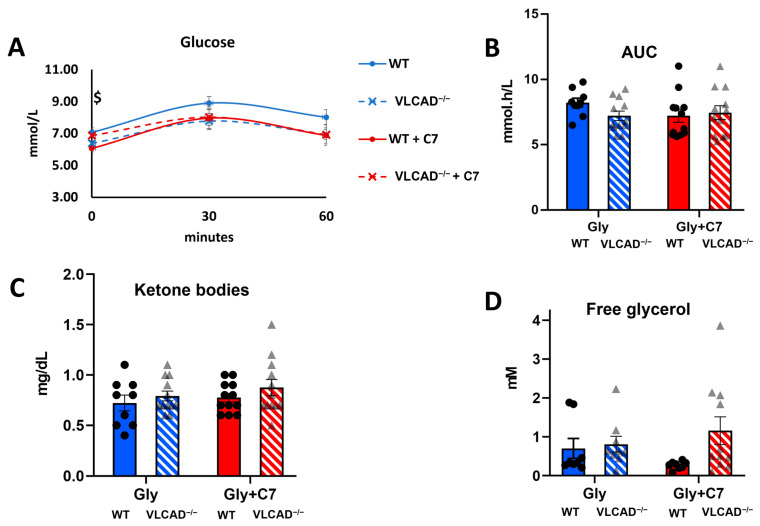
Blood measurement in VLCAD^−/−^ mice with heptanoate (C7). (**A**). Time course of blood glucose concentration. (**B**). Area under the curve (AUC) of blood glucose concentration. (**C**). Ketone body concentration at time point 0. (**D**). Free glycerol concentration in serum at 60 min after bolus injection. Values are mean ± s.e.m., *n* = 10–12. Mice groups consisted of both sexes. Significance differences were measured using multiple unpaired Student’s *t*-tests with Holm–Sidak’s method. ^$^ adjusted *p*-value (*q*) < 0.05 comparison between control and C7 in WT mice.

**Figure 3 nutrients-15-04689-f003:**
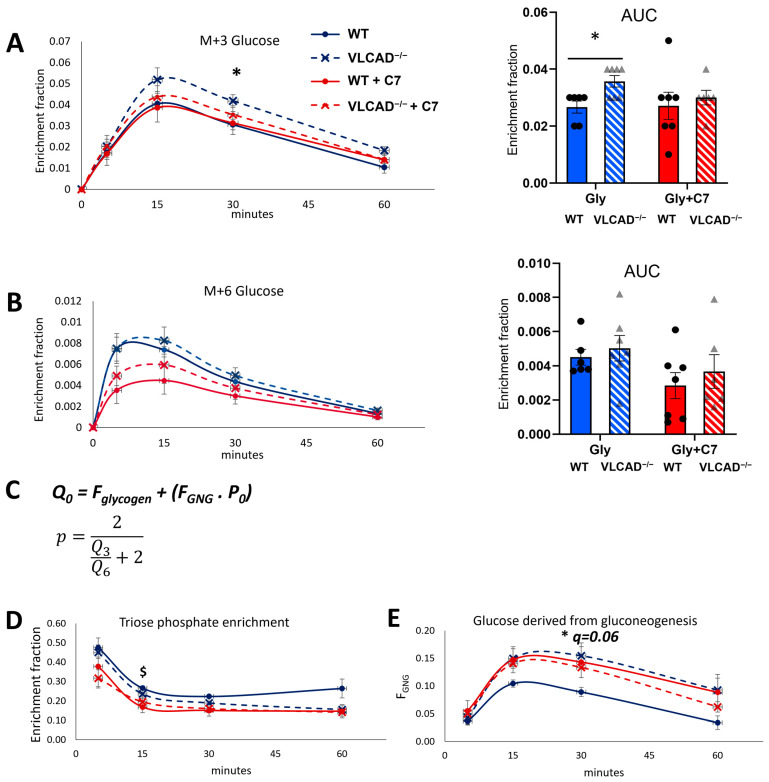
^13^C_3_-glycerol incorporation into blood glucose. (**A**,**B**) Time course and AUC of M+3 and M+6 fractional enrichment in blood glucose. (**C**) The equation to estimate the fraction of glucose derived from gluconeogenesis (FGNG), that is derived from the fractional enrichment of the triose phosphate pool (*p*). *Q*_0_ denotes measured M+0 fraction in blood glucose (see Methods for details). (**D**) The fractional enrichment *p* of the gluconeogenic precursor pool (hepatic triose phosphate). (**E**) The fraction of blood glucose derived from gluconeogenesis. Values are mean ± s.e.m., *n* = 6–7. Significance differences were measured using multiple unpaired Student’s *t*-tests with Holm–Sidak’s method. ^$^ adjusted *p*-value (*q*) < 0.05 comparison between glycerol and glycerol+C7 in WT mice. * *q* < 0.05 comparison between WT and VLCAD^−/−^ mice in control group. Mice groups consisted of both sexes.

**Figure 4 nutrients-15-04689-f004:**
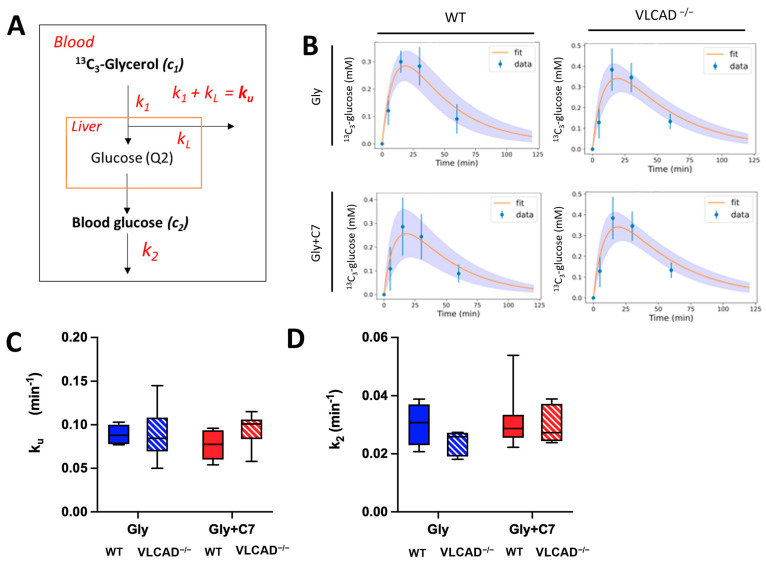
Rate constants (*k*) of glycerol utilization (*k_u_*) and glucose utilization (*k*_2_). (**A**) Schematic representation of used computational modeling to quantify rate constants (see Methods for details). (**B**) The time course of curve fits of ^13^C incorporation in blood glucose from ^13^C_3_-glycerol. Kinetic constant of glycerol utilization. The values of measured data (blue) are mean ± sd. The average and sd of the fitted curved are represented by orange line and purple area, respectively. (**C**) and glucose clearance (**D**) obtained from computational modeling. Values are mean ± s.e.m., *n* = 6–7. Mice groups consisted of both sexes.

**Figure 5 nutrients-15-04689-f005:**
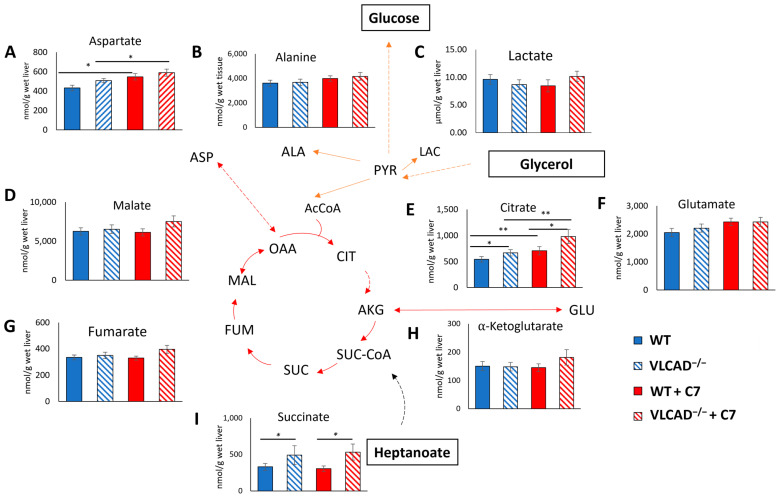
Loss of VLCAD affects metabolic intermediate concentration in the liver. (**A**) Aspartate. (**B**) Alanine. (**C**) Lactate. (**D**) Malate. (**E**) Citrate. (**F**) Glutamate. (**G**) Fumarate. (**H**) *α* -Ketoglutarate. (**I**) Succinate. Values are mean ± s.e.m., *n* = 10–12. Mice groups consisted of both sexes. Significant differences were measured using two-way ANOVA test; * adjusted *p* value (*q*) < 0.05, ** *q* < 0.01.

**Figure 6 nutrients-15-04689-f006:**
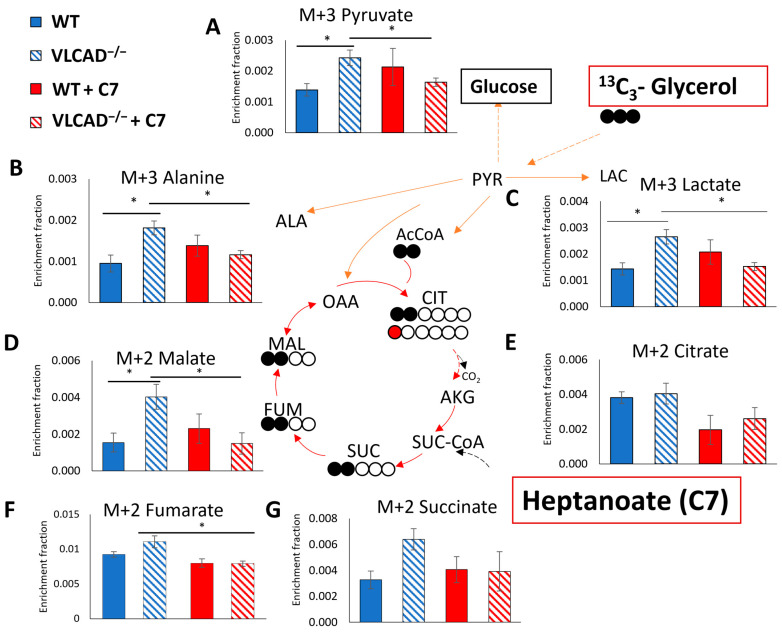
The ^13^C fractional enrichment in liver gluconeogenic substrate of M+3 pyruvate (**A**), M+3 alanine (**B**), M+3 lactate (**C**), M+2 malate (**D**), M+2 citrate (**E**), M+2 fumarate (**F**), and M+2 succinate (**G**). Filled and unfilled circles illustrate the labeled and unlabeled carbon atoms, respectively. Values are mean ± s.e.m., *n* = 10–12. Mice groups consisted of both sexes. Significant differences were measured using two-way ANOVA test; * adjusted *p* value (*q*) < 0.05.

## Data Availability

The data presented in this study are available on request from the corresponding author. The data are not publicly available as they also contain original nonpublished data.

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
