# Peer review of "Heptanoate Improves Compensatory Mechanism of Glucose Homeostasis in Mitochondrial Long-Chain Fatty Acid Oxidation Defect"

_nutrients, 2023, doi:10.3390/nu15214689_

Round 1

Reviewer 1 Report

Comments and Suggestions for Authors

This is an interesting manuscript looking at gluconeogenesis in a murine model of VLCADD. The study is well-designed and executed. It adds to the overall body of knowledge about gluconeogenesis in lc-FAODs. The study was conducted after a 6 hr fast and there was no difference in blood glucose among groups (mild stress). The authors point this out in the 2nd paragraph of the discussion but it would be helpful for context to also mention the results might be considerable different under more stressful conditions such as prolonged fasting/cold exposure or exercise exhaustion when VLCAD-/- exhibit hypoglycemia. It is very surprising to see higher TCA intermediates in the VCLCAD-/- liver. The triheptanoin hypothesis suggests patients with LC-FAODs have lower TCA intermediates – not higher. Again, the authors address this in the discussion line 423-440 but it would be helpful for context to mention this is opposite of the original hypothesis for triheptanoin. I think it also important to have clear language in this paragraph and state “One of the surprising results is that VLCAD-/- mice have higher concentrations of TCA cycle intermediates in the liver.” The discussion then becomes a bit long and hard for the reader to fully understand the implications of the study. The following paragraph (lines 441-456) is hard to understand. It would be better to re-write this section and discuss potential implications of altered gluconeogenesis in VLCADD. If heptanoate provides a gluconeogenic precursor – could fasting times be prolonged before hypoglycemia develops? What about the impact on gluconeogenesis? Is C7 stored in glycogen and that is why it might prevent hypoglycemia? The conclusion should also be more clear about the key results and implications.  

minor comments;

Often associated with medium-chain FAO so would avoid using that abbreviation. 

Line 78 – “ as an alter-“

Lin 244 – glucose using (delte a) computational modelling (Figure 1B).

Path for labeled pyruvate from glycerol not clear

Fig 3E: text suggests difference in glucose from gluconeogenesis different between wt/vlcad but c7 difference disappears – looks like that is d/t change wt group; not vlcad group

Line 315 – may be associated with a significant down regulated (delete of) liver glycogen synthetase

Supplemental figures labeled differently than manuscript – change ko to VLCAD-/- for consistency

Comments on the Quality of English Language

Minor english errors mentioned above but generally good.

Author Response

Thank you very much for your effort and time to give us review and feedback.

Please find in the attached file for the response.

-----------------------------------

Response to Reviewer I

This is an interesting manuscript looking at gluconeogenesis in a murine model of VLCADD. The study is well-designed and executed. It adds to the overall body of knowledge about gluconeogenesis in lc-FAODs. The study was conducted after a 6 hr fast and there was no difference in blood glucose among groups (mild stress). The authors point this out in the 2nd paragraph of the discussion but it would be helpful for context to also mention the results might be considerable different under more stressful conditions such as prolonged fasting/cold exposure or exercise exhaustion when VLCAD-/- exhibit hypoglycemia.

Response: Thank you very much for your kind words and review. We agree with the reviewers that this sentence does not clarify that our results will be different in an experimental setup after exposure to more stressful conditions. We have changed the text in accordance with the reviewer’s suggestions as follows:

We showed that VLCAD-/- mice maintained glucose homeostasis after 6 hours of fasting and showed a higher proportion of EGP from gluconeogenesis than WT mice, strongly suggestive of a compensatory mechanism of glucose homeostasis due to the mitochondrial lc-FAO defect. However, this adaptive mechanism is not able to cope with the energy requirements during intensive stress such as overnight fasting alone or prolonged fasting associated to cold exposure which leads to severe hypoglycemia in different murine models of VLCAD deficiency [28,29] (line 411-421).

It is very surprising to see higher TCA intermediates in the VCLCAD-/- liver. The triheptanoin hypothesis suggests patients with LC-FAODs have lower TCA intermediates – not higher. Again, the authors address this in the discussion line 423-440 but it would be helpful for context to mention this is opposite of the original hypothesis for triheptanoin. I think it also important to have clear language in this paragraph and state “One of the surprising results is that VLCAD-/- mice have higher concentrations of TCA cycle intermediates in the liver.”

Response: We agree and changed the text in accordance to the reviewer’s suggestions. (Line 433-434)

The discussion then becomes a bit long and hard for the reader to fully understand the implications of the study. The following paragraph (lines 441-456) is hard to understand. It would be better to re-write this section and discuss potential implications of altered gluconeogenesis in VLCADD. If heptanoate provides a gluconeogenic precursor – could fasting times be prolonged before hypoglycemia develops? What about the impact on gluconeogenesis? Is C7 stored in glycogen and that is why it might prevent hypoglycemia?

Response: We agree and changed the text to meet the suggestions of the reviewer as follows:

Heptanoate in the form of triheptanoin is already applied in the treatment of  VLCAD deficiency [14]. In patients with lc-FAOD, triheptanoin has been demonstrated to reduce the number of hospitalization and hypoglycemic crisis [14,36]. In VLCAD-/- mice triheptanoin significantly increased liver glycogen in both WT and VLCAD-/- mice compared with normal-chow diet (unpublished data from S. Tucci). We suppose that a triheptanoin based diet provide glycogen precursors through the formation of glucose 6-phosphate [37]. In humans, liver glycogen is used as energy source during prolonged endurance exercise [38]. Therefore, supplementation with triheptanoin diet in situations of high catabolic stress such as prolonged fasting may prevent or at least reduce hypoglycemic events in patients with VLCAD deficiency. (Line 450-458)

The conclusion should also be more clear about the key results and implications.  

Response: We revised the conclusion as follow (Line 491-501)

In summary, our animal study provides an insight into the possible mechanism of triheptanoin for maintaining glucose homeostasis during dietary management in patients with lc-FAO disorder. In VLCAD-/- mice the glycerol supplemented with diet, as component of triheptanoin oil, is used as gluconeogenic substrate to maintain glucose homeostasis. However, heptanoate also acts as a gluconeogenic substrate as it competes with glycerol by reducing its contribution in endogenous glucose production. In addition, heptanoate seems also to be able to improve TCA efficiency.

Further studies with appropriate combinations and concentrations of isotope tracers are necessary to elucidate whether heptanoate provides additional acetyl-CoA (allosteric activator) or propionyl-CoA (anaplerotic substrate).

Often associated with medium-chain FAO so would avoid using that abbreviation. 

Line 78 – “ as an alter-“

Lin 244 – glucose using (delte a) computational modelling (Figure 1B).

Path for labeled pyruvate from glycerol not clear

Response: We changed the text as suggested.

Fig 3E: text suggests difference in glucose from gluconeogenesis different between wt/vlcad but c7 difference disappears – looks like that is d/t change wt group; not vlcad group

Response: We agree. We clarified the sentence as follows:

Without the C7, the fraction of glucose produced by gluconeogenesis was 66% higher in VLCAD-/- mice than in WT mice (Figure 3E, q=0.06). Upon C7 supplementation, this difference disappeared, due to an increase of the contribution of gluconeogenesis in the WT. (Line 314-317)

Line 315 – may be associated with a significant down regulated (delete of) liver glycogen synthetase

Response: We agree and replaced the original sentence by:

… may be associated with a significant downregulation of liver glycogen synthetase. (Line 320-321)

Supplemental figures labeled differently than manuscript – change ko to VLCAD-/- for consistency

Response: We addressed this issue and revised the labels in Supplemental figures as suggested.

Reviewer 2 Report

Comments and Suggestions for Authors

​In this study, an effort was made to show that a deficiency in mitochondrial LC-FAO increases the contribution of the gluconeogenic substrate glycerol to maintain glucose homeostasis. Although this is a popular subject of high research value and falls within the scope of the journal, there are certain issues that the authors need to address specifically.

Lines 81:

Were single-sex mice used in this experiment? Describe this explicitly in Section 2.1 and explain the reason for this setting in the response letter. It should be noted that the effect of the sex of the mouse on the experimental results is not neglected.

Lines 84:

Why is blood sugar measured for only 0-60 minutes?  Why not measure blood sugar for a longer period of time (say 120 minutes, or even 240 minutes)? What's the reason?

Isotope tracers are a good method for nutritional studies. Did the authors consider collecting animal urine and feces in this paper to better identify metabolic pathways?

Lines 201-203:

Why and how did the authors simplify the system of ODEs models?

Lines 379-380:

Personally, I would like to know how the data mentioned there supports the previous data in this study that VLCAD-/- mice have rapidly elevated metabolism resulting in 13C enrichment dilution in TCA cycle intermediates. ​Please explain that.

In the Introduction section, the authors claim to have constructed a computational model to quantify glucose clearance in VLCAD-/- mice. Please explain how and which aspects of the evaluation presented in this paper reflect the validity and reliability of the computational model.

In summary, I recommend that the article be further evaluated after major revision.

Comments on the Quality of English Language

Minor editing of English language required.

Author Response

Thank you very much for your effort and time to give us review and feedback.

Please find in the attached file for the response.

---------------------

Response to Reviewer II

In this study, an effort was made to show that a deficiency in mitochondrial LC-FAO increases the contribution of the gluconeogenic substrate glycerol to maintain glucose homeostasis. Although this is a popular subject of high research value and falls within the scope of the journal, there are certain issues that the authors need to address specifically.

 Lines 81:

Were single-sex mice used in this experiment? Describe this explicitly in Section 2.1 and explain the reason for this setting in the response letter. It should be noted that the effect of the sex of the mouse on the experimental results is not neglected.

Response: Thank you very much for your kind words and review. In our experimental setting we used both sexes, male and female, in equal numbers. This point is described in the section “Material and Methods” Section 2.1 lines 86-87:

Each group consisted of 10-12 mice, with equal numbers of male and female mice.

We fully agree that sex play role in lipid metabolism and in response to dietary supplementation. Indeed, previous studies of our group have focused on the sex specific development of severe metabolic syndrome in female VLCAD-/- mice upon MCT diet in the long run whereas male VLCAD-/- mice were protected.

Here, we could not observe a sex specific response to the labelled substrates but only a significant genotype effect, independently of the sex of the mice. Therefore, the results from male and female mice are represented as a single group (WT vs VLCAD-/-). Sexual dimorphism in response to triheptanoin has been already described in VLCAD-/- mice after a long term (12 months) supplementation with a specific diet. We cannot exclude that sex specific differences may appear at a later time point, however, the period investigated in this study (90 minutes) is too short. 

To avoid misunderstanding regarding the sex composition of the mice groups we added following description in each Figure legend: “Mice groups consisted of both sexes” 

Lines 84:

Why is blood sugar measured for only 0-60 minutes?  Why not measure blood sugar for a longer period of time (say 120 minutes, or even 240 minutes)? What's the reason?

Response: From Fig. 4B it is clear that most of the dynamics happens in the first hour and was sufficient to fit the data. Nevertheless, a 2h timepoint would have been informative to capture the full dynamics. We added a sentence in section 3.4 (Line 337-341):

As shown in Figure 4B, the model achieved a good fit of the measured data in all experimental groups. The period of 60 min during which label incorporation was followed, was sufficient to cover most of the dynamics. Yet, an additional 120 min timepoint could have been informative.

Isotope tracers are a good method for nutritional studies. Did the authors consider collecting animal urine and feces in this paper to better identify metabolic pathways?

No, we did not consider this, since we do not see how it would have contributed to solving our research questions.

Lines 201-203:

Why and how did the authors simplify the system of ODEs models?

Response: The original model by Vieira Lara et al. was developed for an oral glucose tolerance test with label. Here we studied the kinetics of labelled glucose originating from labelled glycerol. We clarified this as follows. (Line 203-207)

To determine the apparent kinetic constants of blood glucose turnover, we adapted the system of ordinary differential equations (ODEs) model described by Viera-Lara [25] for our experimental setup. The original model was developed for an oral glucose tolerance test with label. In the present study the label comes from the gluconeogenic conversion of 13C3-glycerol into glucose.

Lines 379-380:

Personally, I would like to know how the data mentioned there supports the previous data in this study that VLCAD-/- mice have rapidly elevated metabolism resulting in 13C enrichment dilution in TCA cycle intermediates. ​Please explain that.

Response: We assume that the reviewer refers to the m+1 labelling in citrate in line 376-377. This follows the same pattern as the other metabolites, namely an elevated label in the VLCAD-/- mice, which was brought back to WT level by triheptanoate. We added a sentence (line: 385-387)

We showed that both the genotype and the C7 bolus affected the m+1 citrate enrichment (Figure S1). The pattern was qualitatively the same as that of m+2 TCA cycle intermediates, i.e. elevated label incorporation in the VLCAD-/- group, and normalisation to WT level when C7 was given.

In the Introduction section, the authors claim to have constructed a computational model to quantify glucose clearance in VLCAD-/- mice. Please explain how and which aspects of the evaluation presented in this paper reflect the validity and reliability of the computational model.

Response: Validity and reliability of a model is a difficult question. Obviously, any model is only a simplified representation of the biological reality. However, the fact that the model describes the data well, indicates that the model is a good representation of the glucose production and clearance. We clarified this in the text as follows (line: 337-340)

As shown in Figure 4B, the model fitted the measured data well in all experimental groups, demonstrating that it is a good representation of the system.

Round 2

Reviewer 2 Report

Comments and Suggestions for Authors

All responses except the last one are acceptable.

Regarding my last concern, the author failed to provide me with a satisfactory answer. As stated by the authors, the models mentioned in the paper only explain the relevant data well in this experimental scenario, and no evidence has been presented to show the validity and reliability of the computational models mentioned there. Therefore, I personally recommend the authors to revise the corresponding introduction and conclusion sections to make the paper more rigorous and thus avoid reader misunderstandings.

Comments on the Quality of English Language

Minor editing of English language required.

Author Response

Response to Reviewer II (Round 2)

All responses except the last one are acceptable.

Regarding my last concern, the author failed to provide me with a satisfactory answer. As stated by the authors, the models mentioned in the paper only explain the relevant data well in this experimental scenario, and no evidence has been presented to show the validity and reliability of the computational models mentioned there. Therefore, I personally recommend the authors to revise the corresponding introduction and conclusion sections to make the paper more rigorous and thus avoid reader misunderstandings.

Thank you very much for the comments. We understand the concern. A goodness-of-fit is a way to evaluate the computational model of enzyme kinetics. We have now added some further explanation about the origin of the model in the Methods section. Moreover, as this type of model was originally developed and validated for oral glucose tests, we emphasise in the Discussion that the main conclusion (the unchanged k2, representing the peripheral glucose consumption) is overlapping between the different models. We hope that this solves the concern. We added the following text (red colour):

Methods (Line:203-205):

To determine the apparent kinetic constants of blood glucose turnover, we adapted the system of ordinary differential equations (ODEs) model described by Viera-Lara [23] for our experimental setup. The original model was developed for an oral glucose tolerance test with label, building on earlier models by Dalla Man et al [25], and validated for both humans and mice [23,26]. In the present study, the label comes from gluconeogenic conversion of 13C3-glycerol into glucose.

Result (Line: 336-341):

To evaluate adequacy of the computational model, the measured data should be consistent with the model [28]. We fitted the time course of the m+3 glucose data to this model (Eq. 17). As shown in Figure 4B, the model (orange line) fitted the measured data (blue dot) well in all experimental groups, demonstrating that is a good representation of the system. The period of 60 minutes during which label incorporation was followed, was sufficient to cover most of dynamics. Yet, an additional 120 min timepoint could have been informative.

Discussion (Line: 336-342):

The computational model applied in this study indicates that high dependence on glucose in peripheral tissues of VLCAD-/- mice [32] may not play the main role as compensatory mechanism of glucose homeostasis, as illustrated by the rate constant k2, which remained identical in all groups. We emphasize that we adapted a model that was originally constructed and validated for oral glucose tolerance tests [23]. However, the difference between the models resided only in the production/absorption of glucose, not in the consumption by peripheral tissues, represented by k2, in which we are interested here.
